# Construction of Porous Starch-Based Hydrogel via Regulating the Ratio of Amylopectin/Amylose for Enhanced Water-Retention

**DOI:** 10.3390/molecules26133999

**Published:** 2021-06-30

**Authors:** Huiyuan Luo, Fuping Dong, Qian Wang, Yihang Li, Yuzhu Xiong

**Affiliations:** Department of Polymer Materials and Engineering, College of Materials and Metallurgy, Guizhou University, Guiyang 550025, China; gs.luohy17@gzu.edu.cn (H.L.); fpdong@gzu.edu.cn (F.D.); gs.qingwang19@gzu.edu.cn (Q.W.); gs.liyh18@gzu.edu.cn (Y.L.)

**Keywords:** ultrasonic chemistry, hydrogel, starch, porous structure, water-retaining

## Abstract

The performance of hydrogels prepared with traditional natural starch as raw materials is considerable; the fixed ratio of amylose/amylopectin significantly limits the improvement of hydrogel structure and performance. In this paper, starch hydrogels were prepared by physical blending and chemical grafting, with the aid of ultrasonic heating. The effects of different amylose/amylopectin ratios on the microstructure and water retention properties of starch hydrogels were studied. The results show that an increase in amylopectin content is beneficial to improve the grafting ratio of acrylamide (AM). The interaction between the AM grafted on amylopectin and amylose molecules through hydrogen bonding increases the pores of the gel network and thins the pore walls. When the amylopectin content was 70%, the water absorption (swelling 45.25 times) and water retention performance (16 days water retention rate 44.17%) were optimal. This study provides new insights into the preparation of starch-based hydrogels with excellent physical and chemical properties.

## 1. Introduction

Global warming conditions have resulted in water shortages globally [1], leading to serious desertification. Water loss and unfixed water are the main causes of desertification, and the development of water-absorbing and water-retaining materials suitable for desertification is one of the serious challenges facing scientists [2,3,4,5]. High-efficiency water storage materials, such as water-absorbing and water-retaining gels, have become popular research interests. Since Wichterle and Lim first reported the application of hydrogel materials in contact lenses in 1960, hydrogels have been widely used in tissue engineering, controlled drug delivery, agriculture, bioactive substance protection and water purification [6,7,8,9,10].

The unique three-dimensional network structure of the absorbent hydrogels has hydrophilic functional groups, which can absorb a large amount of water [11]. Its excellent water storage and excellent ability to slowly release water from swollen hydrogels have been widely used in various industries, such as agriculture and horticulture [12,13,14,15,16]. In water-scarce areas, the use of absorbent hydrogels can help reduce irrigation water and maintain soil moisture. Studies have shown that absorbent hydrogels can regulate the evaporation and infiltration of water by affecting the density and structure of the soil, thereby significantly improving soil nutrient retention and physical properties [17]. Absorptive hydrogels can also suppress soil compaction trends by reducing the number of irrigations and preventing soil erosion [12]. Its ability to improve microbial activity brings great benefits to agricultural production [18].

Intermolecular interactions, covalent and non-covalent interactions in the production of absorbent hydrogels have been extensively studied [19,20,21]. Most absorbent hydrogels on the market have several disadvantages, including high price, poor degradability, and numerous environmental problems caused by long-term use [22,23]. Hydrogels made from natural ingredients, such as sodium alginate, gelatin, cellulose, chitosan, and starch [24,25,26,27,28,29,30,31,32,33], have received more and more attention in recent years due to their safety, biocompatibility, and biodegradability [34]. Amorphous and crystalline regions make up the multi-scaled structure of starch. The amorphous area is filled with branch points of amylopectin and single helix molecules of amylose, and the crystalline area is composed of a double helix molecular structure of amylose [35]. Starches with different amylose/amylopectin content have different phase transformation behaviors [36,37] and rheological properties [38,39]. In addition, studies have shown that the shear degradation of corn starch, with different amylose content, is attributed to the greater sensitivity of the rigid crystal structure of amylose to shear stress, compared to the flexible amorphous structure of amylopectin [40]. Rath et al. used graft copolymerization of acrylamide with amylose corn-starch, common corn-starch and amylopectin corn-starch to produce starch-based superabsorbent resin. The results showed that the side chain points of amylopectin-based superabsorbent resin are few and long. Straight chain starch-based superabsorbent resin has many short side chain points, and amylopectin-based superabsorbent resin shows a stronger flocculation effect [41]. Research by Castel et al. showed that the type of starch has a significant effect on the grafting degree. Starch containing a high amylopectin content makes it easier to graft more acrylonitrile side chains [42]. Starch with high amylose content reduces the grafting ratio of acrylonitrile [43]. However, the influence of amylose/amylopectin on the microstructure of starch-based gels has not been studied in depth.

As a simple and environmentally friendly method, it has been proved that the preparation of acrylamide-modified polysaccharides by ultrasound-assisted methods can shorten the graft copolymerization time and improve the grafting efficiency [44]. In this study, the synergistic effect of amylose/amylopectin on the microstructure, thermal stability and water absorption and retention properties of starch-based gels prepared with ultrasonic heating was investigated by physical mixing method. The information obtained from this study facilitates the understanding of the synergistic mechanism of amylose/branched chain molecules on the starch gel structure. Moreover, we aimed to design a new type of starch-based hydrogel with a precisely controlled structure. This degradable starch-based gel has broad application prospects in agriculture, as well as the mitigation of desertification.

## 2. Results and Discussion

### 2.1. Potential Mechanism of Amylose/Amylopectin Synergistic Regulation of Gel Microstructure Interaction

Unlike natural starch, in which amylose is encapsulated in granules, in this study, amylose and amylopectin were physically blended, and distilled water was added to make the amylose and amylopectin evenly mixed under the action of ultrasonic heating (Power:400 W Frequency:40 KHz). As the temperature increased, the amylopectin granules began to absorb water and swell, while the amylose molecules with a helical structure were heated and stretched. Acrylamide was added when the mixed starch gelatinization was complete. Under the action of ultrasonic heating, energy was introduced in a short time, and a large number of micro bubbles were generated. The bubbles burst in a few microseconds and provided additional energy for the entire reaction system [45,46,47,48,49,50], so that AM is evenly dispersed in the starch solution and tends to graft on amylopectin with more branch points.

As the reaction progressed, the grafted AM molecular chain and amylose molecular chain formed a network structure through the interaction of hydrogen bonds (as shown in Figure 1), similar to the hydrogen bond between starch and polysaccharide molecules [51]. When the content of amylopectin was low, although the viscosity of the reaction system is low so that the amylose molecules can be stretched well, the starch material cannot form a regular three-dimensional gel or three-dimensional network structure, due to the low grafting ratio and insufficient cross-linking degree. With the increase in amylopectin and the increase in branching points, AM can be more easily grafted, and with a higher grafting degree it can form stronger hydrogen bonds with amylose molecules, resulting in a regular three-dimensional gel network structure. When the amylopectin ratio was 70% (AP70), the optimal pore structure in the gel was formed. With the increase in amylopectin, the relative content of amylose decreased, while the grafting ratio and crosslinking ratio increased. The increase in amylopectin content led to an increase in viscosity and the viscosity of the reaction system increased, which affected the extension of amylose. Therefore, the backbone structure did not support the formation of more amylopectin/AM molecules, resulting in the collapse of the gel network.

### 2.2. Structural Characteristics of Starch Gel

#### 2.2.1. Chemical Structure Analysis

In Figure 2, the FTIR spectra of AM, AP0 (starch gel) and amylose (amylose) and amylopectin (amylopectin) are shown. It can be seen from Figure 2 that for amylopectin, there is a significant peak of the stretching vibration of the O-H group of the glucose unit of the starch molecular chain at 3440 cm^−1^ and stretching of the methylene group on the glucose unit at 2930 cm^−1^ [18]. The vibration peak, 1643 cm^−1^, is the amorphous peak of the water adsorbed in the amorphous region of amylose. Peaks at 1160 cm^−1^, 1080 cm^−1^, and 1016 cm^−1^ are attributed to the triplet peak of C-O-C stretching of the glucose ring on the starch molecular chain [18]. For AP0, the broad peak at 3426 cm^−1^ is generated by the superposition of stretching vibration peaks of -OH in starch and -NH_2_ in AM, while the absorption peak at 2980 cm^−1^ is due to the offset of stretching vibration of methylene caused by the access of AM. At 1668 cm^−1^, the absorption peak caused by the superposition of the -C=O stretching vibration peak on the amide and the N-H bending vibration peak of the primary amide can be clearly seen [52]. The formation of a peak at 1384 cm^−1^ indicates that there is C-N stretching vibration. In addition, the absorption peaks at 1153 cm^−1^, 1088 cm^−1^, and 1048 cm^−1^ indicate the integrity of the starch glucose ring structure [52]. Based on the analysis of the above peaks, it can be seen that AM is successfully grafted onto starch.

#### 2.2.2. Analysis of Graft Ratio of Starch Gel

According to Figure 3, with the increase in the amylopectin ratio, the grafted ratio of gel samples showed an overall increasing trend, indicating that the increase in amylopectin could improve the graft ratio of gel samples. Analysis shows that the amylopectin chain was disordered during the gelation process, and amylopectin had more branch points, which provided more chemical grafting points [41,42,43,44]. Therefore, the grafting ratio of starch gels increased with the increasing amylopectin content. However, the decrease in the grafting ratio of AP50 may be due to the fact that amylose molecules and amylopectin molecules are more likely to form hydrogen bonds at this time, which hinders the grafting of AM on the starch molecular chain [53]. The grafting ratio of the starch hydrogels roughly follows its crystallinity.

#### 2.2.3. Crystallization Performance Analysis

As shown in Figure 4, in the hydrogel diffraction pattern, when the amylopectin content is 0 (AP0) there is almost no diffraction peak in the hydrogel sample, and strong diffraction occurs when the amylopectin content is 10% (AP10). The positions of the peaks are 2θ = 16°, 28.6°, 40°, and 49°. Due to the addition of amylopectin, the outer amylopectin of the amylopectin participates in the crystallization process of the gel, which strengthens the short-term retrogradation process of amylose and subsequently enhances crystallinity [54]. As the amylopectin content (AP30) increases, the crystallization peak of the hydrogel gradually weakens. The analysis suggests that the increase in amylopectin may increase the grafting ratio of the gel sample during the reaction, and the decrease in amylose content slows the starch retrogradation [55], making the crystallinity of the prepared gel weak. When the mass content of amylopectin increases to 50% (AP50), the diffraction peak of the hydrogel suddenly increases. The analysis suggests that when the amylopectin content is nearly equal to the amylose content, the amylopectin molecule may be more likely to interact with the amylose, forming hydrogen bonds between molecules and reducing the reaction of acrylamide and amylopectin to some extent [56]. Then, in the rejuvenation stage of the starch gel, with the extension of the cooling time, the uncrossed linked amylose and amylopectin molecular chains can be rearranged by double-helix winding [57], which increases the crystallinity of the gel. With the further increase in amylopectin content, the crystallization peak of the gel weakened, and the crystallization peak of the gel sample AP70 completely disappeared. This indicates the hydrogel became an amorphous structure. Therefore, it is preliminarily believed that the increase in amylopectin may continue to increase the starch grafting rate and increase the ratio of gel crosslinking, thereby affecting the rearrangement of the amylose/amylopectin molecular chain regeneration stage.

#### 2.2.4. Thermogravimetric Analysis (TGA)

As shown in the degradation curve of amylose/amylopectin in Figure 5a, there are two significant mass losses in the temperature range of 50–100 °C and 300–350 °C. The former is the evaporation peak of water in amylose/amylopectin, while the latter is the decomposition and degradation peak of amylose/amylopectin [5]. The degradation curves of gel samples with different amylopectin contents indicate that there are three more obvious temperature ranges for gel samples: 50–100 °C, 250–300 °C, and 350–450 °C. These ranges correspond to the sample water evaporation, degradation of starch molecular chain, and degradation of AM graft chain, respectively [58]. The thermal degradation temperature of starch molecular chain of gel samples was lower than that of amylose/amylopectin. The analysis indicates that because of the access of AM, the ordered structure of starch in the gelatinization process was destroyed. Therefore, it was easier to degrade. The peak of degradation loss at 350–450 °C indicates that the side chain of the grafted polyacrylamide may be decomposed or broken [59]. As shown in Figure 5a,b, by comparing the degradation curves of amylose/amylopectin and starch gel samples, it can be found that the gel samples with acrylamide graft cross-linking reaction have a better thermal stability. Ultimately, there were still some undegraded residues in the gel samples, and the most residue is 32%.

#### 2.2.5. SEM Characterization of Starch Gel Micromorphology

Figure 6 shows the porous structure of starch gel samples. When the amylopectin content was 0 and 10% (AP0, AP10), the gel pores were sparse and had no stereoscopic structure. When the content of amylopectin increased, the porous structure of the hydrogel became more regular.

Additionally, the pore wall thickness gradually decreased, and the pore size increased. This is because the short linear chain formed by acrylamide-grafted amylopectin is easy to entangle and significantly promoted the construction of a network structure [60]. With the increase in grafting degree, the molecular chain of AM on the branched starch increased and the gel hole increased. The reason for the thinning of the hole wall with the increase in amylopectin ratio is that a higher grafting ratio enhances the interaction between AM grafted chains and amylose molecular chains. Therefore, fewer molecular chains can form the pore wall of the gel network, as shown in Figure 6c,d. Moreover, the increase of amylopectin can increase the crosslinking ratio of the gel, resulting in higher porosity [61]. At the same time, the relative content of amylose, which is more conducive to the formation of three-dimensional network skeleton structure, decreases, and the pore wall becomes thinner. When the content of amylopectin increased to 70% (AP70), the high porosity and the skeleton backbone of the three-dimensional network structure were enough to support the construction of the network structure, thus forming the best network structure. When the amylopectin content continued to increase, the amylose of the skeleton structure further decreased, and the higher cross-linking ratio of the gel inhibited the extension of the amylose molecular chain. As a result, the skeleton of the main chain of the three-dimensional network is insufficient to support the construction of more network structures, and the porous structure of the hydrogel collapsed significantly (Figure 7f).

### 2.3. Water Absorption Properties of Starch Gel

#### 2.3.1. Swelling Performance Analysis

Figure 7 is a water swelling test for different amylopectin contents (0–90%). According to Figure 7, with the increase in amylopectin content, the swelling rate of hydrogels generally increased first and then decreased. When the amylopectin mass ratio as 70% (AP70), the water swelling rate of the gel reached the maximum (45.29 times). The analysis indicates that in the three-dimensional network structure of the starch gel, the network of AM on the amylopectin and the larger network formed by the hydrogen bonding interaction of the amylose and the main chain formed the network balance. The presence of a small network can enhance the capillary action in the microporous structure. This loose network interpenetration structure can improve the ability of starch gel to contain water. The proportion of the small network increased as the content of amylopectin increased, so that the water absorption capacity of the gel was enhanced. However, the space in the gel for containing water was reduced, thus, the water swelling rate as also reduced.

#### 2.3.2. Analysis of Water Retention Performance

Figure 8 is a water retention test chart of gel samples over 16 days. Figure 8 shows that the water loss of gel samples in the first two days was significant (about 20%), and after two days, the water loss was less drastic. According to the swelling theory of Flory-Huggins, the water absorbent gel absorbs water through hydrophilic interaction. After absorbing water, the hydrophilic group on the side chain of the gel is ionized with a uniform charge, which leads to electrostatic repulsion that helps the gel to be stretched to form a three-dimensional network structure [62]. According to Flory’s [62] gel theory, after absorbing water, a part of the gel substance exists as free water in the gel network structure, while the other part is adsorbed by the water-absorbing groups in the gel and exists as bound water in the gel structure. Therefore, under natural conditions, free water volatilizes first, and then the bound water adsorbed on the water-absorbing group desorbs and then volatilizes. It can also be seen from the Figure that when the proportion of amylopectin was 70% (AP70), the water retention of the gel was the best. It can be inferred from the previous analysis that the increase in the small network structure played an important role, making more hydrophilic groups bind to water.

## 3. Materials and Methods

### 3.1. Materials

Amylose and amylopectin were purchased from Anhui Kuer Bioengineering Co., Ltd. (Anhui, China). *N*,*N*′-methylenebisacrylamide (*N*,*N*′-MBA, AR, ≥99%) and acrylamide (AM, AR, ≥99%) were from Aladdin (Shanghai, China). Cerium ammonium nitrate (CAN, AR, ≥99%) was purchased from Sinopharm Group Chemical Reagent Co., Ltd. (Ningbo, China). All chemicals were analytically pure. Sodium hydroxide (NaOH, AR, ≥96.0%) was provided by Yongda Chemical Reagent Co., Ltd. (Tianjin, China).

### 3.2. Gel Synthesis

One gram of starch with different amylopectin quality content ratios (0%, 10%, 30%, 50%, 70%, 90%, respectively) were weighed, and the samples were named AP0, AP10, AP30, AP50, AP70, and AP90, respectively. Three milliliters of distilled water were added to the glass bottle which contained the sample and mixed evenly in the glass bottle. AM (1.5 g) was added to the starch mixture and mixed until uniform. The sample was mixed via ultrasound for 10 min. *N*,*N*′-methylenebisacrylamide (0.1 g) was added to the sample bottle and mixed ultrasonically for 10 min. The ultrasonic temperature was 60 °C. After the temperature had stabilized in 60 ℃, the sample was placed in the ultrasound and stirred for 10 min. The prepared CAN solution (0.03 g/mL) was combined with 1 mL of each sample and allowed to react for 30 min. Then, a certain concentration of sodium hydroxide solution was added, and the saponification reaction continued for 10 min. After repeatedly washing the reaction product with distilled water, 30 mL of distilled water was added to the product, and it was left to swell for 6 h to completely dissolve unreacted substances. After 6 h, it was washed again with water for 2 min. The gel was placed in a lyophilizer for 48 h and set aside.

### 3.3. Characterization of the Gel

#### 3.3.1. Fourier Transform Infrared Spectroscopy (FTIR)

The samples were characterized by Fourier transform infrared spectroscopy (NEXUS6700, Thermo Fisher Scientific, Waltham, MA, USA). The range of the scanning wave number was 500–4000 cm^−1^ at the wavelength of potassium bromide.

#### 3.3.2. XRD Analysis

The crystallinity of the gels was analyzed by X ray diffraction (Xpert Pro MPD, Xpert PANalytical, Almelo, Netherlands). Using Cu Ka as the X ray photoelectron excitation source, the test conditions were as follows: tube flow 100 mA, tube pressure 40 kV, step length was 2θ = 0.02 degree, and 2θ range was 5° to 75°.

#### 3.3.3. TGA Analysis

A PerkinElmer Diamond TGA system (Waltham, MA, USA) was used to determine the thermal decomposition of the gel. The samples were heated from 35°C to 650 °C at 10 °C/min under a nitrogen atmosphere and maintained at 650 °C for 1 min prior to analysis.

#### 3.3.4. SEM Analysis

A scanning electron microscope (JSM-7500F, Akishima City, Tokyo, Japan Electronics Corporation) was used to observe the microscopic morphology of the gel. The samples were gold-sputtered prior to analysis.

#### 3.3.5. Analysis of Grafting Ratio

In the graft copolymerization, the graft efficiency is often used to evaluate the ratio of graft copolymerization. The calculation of grafting efficiency is generally performed by the mass analysis method [63]. A certain amount of crude product is weighed and wrapped in filter paper. It is extracted in a mixed solution of ethylene glycol and glacial acetic acid with a volume ratio of 6:4 for 24 h to remove homopolymerization. The material was then washed with ethanol and dried under vacuum to a constant weight. Grafting efficiency is calculated as follows:(1)$$G=\frac{W_{2}}{W_{1}}\times 100\%$$
where *G* represents the gel grafting degree; *W*_1_-quality of the crude product (g); *W*_2_-quality of the product (g). The results were averaged three times.

#### 3.3.6. Water Absorption Performance Test

##### Water Swelling Rate (Swelling) Test

The water swelling rate (Swelling) was measured using the method studied by Xiao et al. [5]. The dried sample gel (0.5 g) was placed in a bag and immersed in distilled water at an ambient temperature for 6 h. Afterwards, the sample bag was hung to dry until no water droplets were observed. The gel was wiped with absorbent paper towels to remove any free water. The net weight of each sample gel was obtained. The water swelling rate (*S*) is calculated as follows:(2)$$S(g/g)=\frac{\left(M_{1}-M_{2}\right)}{M_{2}}\times 100\%$$
where *M*_1_ and *M*_2_ (*g*) are the weight of the expanded and dried samples, respectively. All results were calculated as the average of three replicates. The results were averaged three times.

##### Water Retention Performance Test

The method published by Olad et al. [64] was used to study the water retention of the sample gel. The gel sample was placed in a plastic cup containing 100 g dry loam (below 20 mesh). After that, 50 mL of distilled water was poured into a plastic cup and weighed (*W*_0_). The plastic cup was kept at room temperature and weighed daily (*W_t_*) over 16 days. Finally, the soil water retention rate (*W*) was determined by the following equation:(3)$$W=\frac{W_{t}-W}{W_{0}-W}\times 100\%$$

The results were averaged three times.

## 4. Conclusions

In this study, amylose/amylopectin with different ratio and acrylamide were utilized as raw materials to prepare water absorbent gels with the aid of ultrasonic heating. The difference in the content of amylopectin determines the grafting ratio of AM and affects the water absorption, as well as the water retention capacity, of the hydrogel. Additionally, the hydrogen bonding between the side chain structure (AM) and the main chain structure (amylose) also influences the water absorption and water retention ability of hydrogels. Acrylamide tends to be grafted onto amylopectin to form a side chain structure and to form more pores of gel network, increasing the ratio of amylopectin. The increased grafting ratio enhanced the interaction between AM and amylose. So, even fewer side chains could build a regular network structure, making the pore wall thinner, and increasing the water absorption and water retention capacity of the gel. However, as the ratio of amylose (the skeleton structure) decreases, the network supporting capacity of the main chain decreases, leading to the collapse of the gel network structure, thereby reducing the water absorption and water retention capacity of the hydrogel. When the amylopectin ratio reaches a certain proportion (70%), the optimal network pores could be formed in the gel structure, and the starch gel has the optimal water absorption and water retention properties. In this work, the synergistic effect of amylose and amylopectin on the formation of starch gel microstructure has been well studied. The water absorbent hydrogel has been developed from natural starch, which has great potential in agricultural environments.

## Figures and Tables

**Figure 1 molecules-26-03999-f001:**
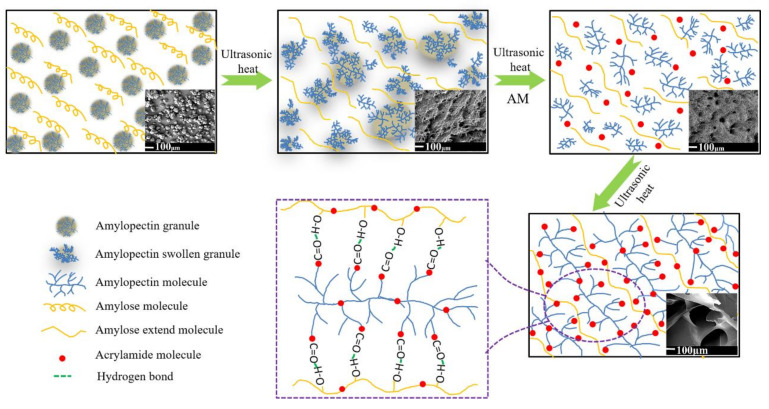
Schematic of amylose/amylopectin synergistic regulation of gel microstructure.

**Figure 2 molecules-26-03999-f002:**
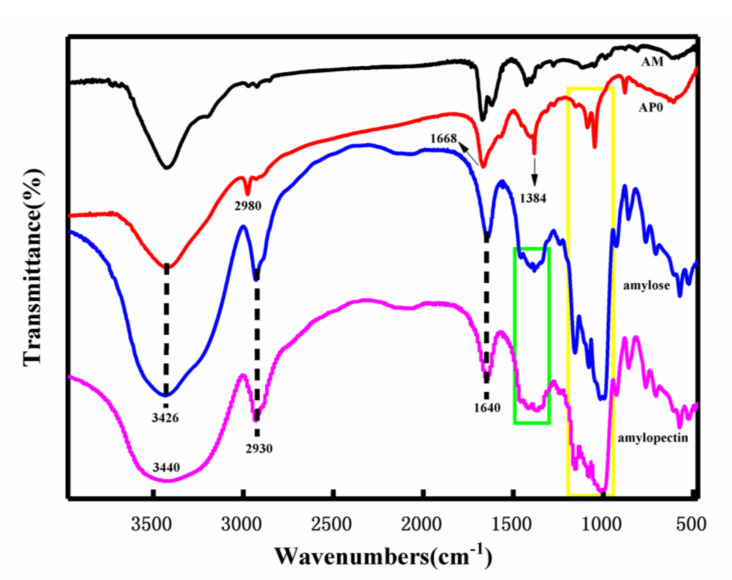
Infrared spectra of acrylamide (AM), starch gel (AP0), amylose, amylopectin.

**Figure 3 molecules-26-03999-f003:**
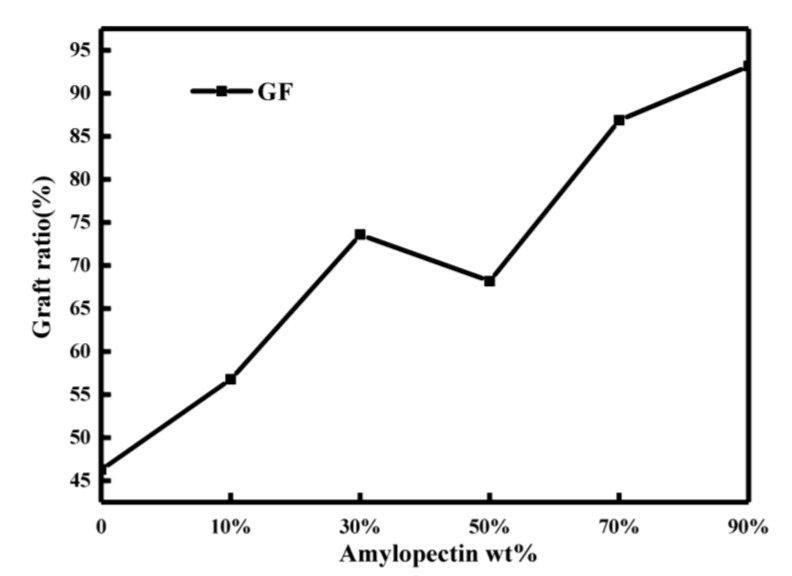
Grafting ratio of gel samples.

**Figure 4 molecules-26-03999-f004:**
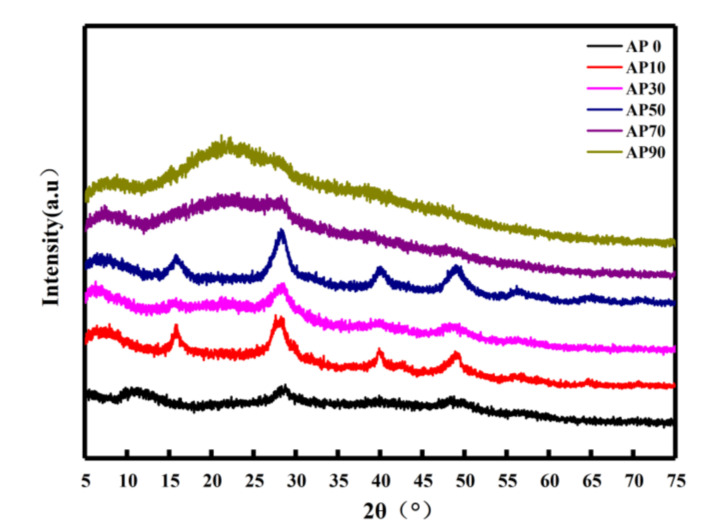
XRD pattern of starch-based hydrogel.

**Figure 5 molecules-26-03999-f005:**
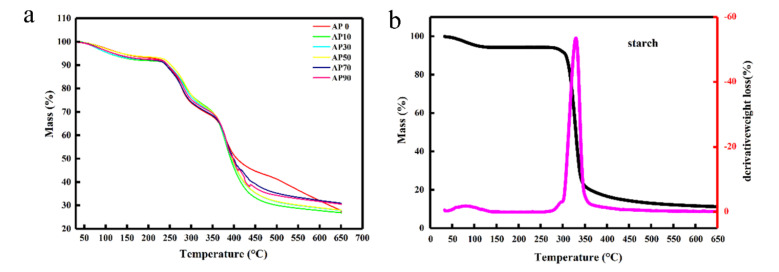
(**a**) TG degradation curves of amylose/amylopectin and starch gel samples and (**b**) TG degradation curves of starch.

**Figure 6 molecules-26-03999-f006:**
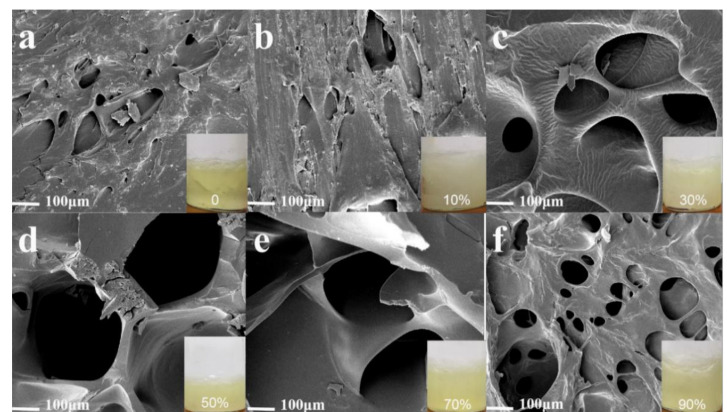
(**a**) SEM image of starch gel with 0% amylopectin content/(**b**) SEM image of starch gel with 10% amylopectin content/(**c**) SEM image of starch gel with 30% amylopectin content/(**d**) SEM image of starch gel with 50% amylopectin content/(**e**) SEM image of starch gel with 70% amylopectin content/(**f**) SEM image of starch gel with 90% amylopectin content.

**Figure 7 molecules-26-03999-f007:**
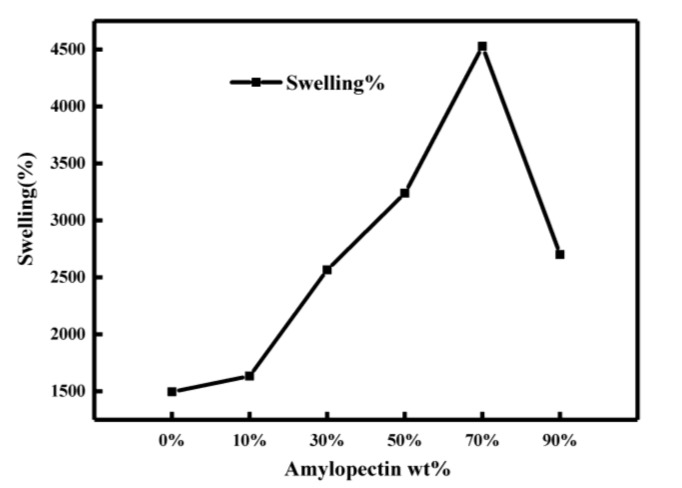
Water swelling diagram of gel sample.

**Figure 8 molecules-26-03999-f008:**
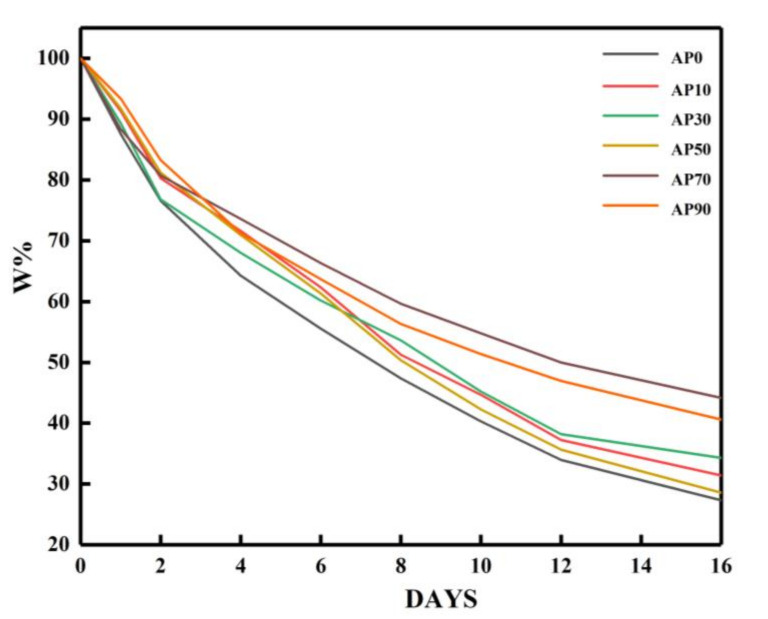
Water retention performance of gel samples.

## Data Availability

All data generated or analyzed during this study are included in this published article.

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
