# Peer review of "Construction of Porous Starch-Based Hydrogel via Regulating the Ratio of Amylopectin/Amylose for Enhanced Water-Retention"

_molecules, 2021, doi:10.3390/molecules26133999_

Round 1
Reviewer 1 Report
The herein presented manuscript gives us a new insight into the preparation of starch-based gels with different amylopectin to amylose ratio. Proposed method utilizes ultrasonic heating to delivery gels with improved, both, physical and chemical properties. Although research has certain level of innovation it has a very serious flaws, both, technical and fundamental. I strongly suggest a major revision of the manuscript before any further considerations regarding publishing. I would also like to address further issues related to the conducted research and its presentation.
Technical issues:
- Some spaces are missing at several places in the manuscript, like: line 50, line 147…
- Certain parts are very confusing and need to be rewritten, as the following one: According to the analysis, due to the addition of amylopectin, the outer amylopectin of the amylopectin…
- Barely anything is visible on the inset figure of the Figure 4.
- Poor choice of the color makes the yellow line at the same graph almost invisible.
- Graphs are put in the manuscript before they are even mentioned in the text, which makes the entire story harder to follow.
Fundamental issues:
- Electrostatic forces between AM and amylopectin are mentioned in the abstract of the manuscript, but not after in the manuscript which brings the confusion about the well-known mechanism of gel formation of these compounds.
- Figure 1 definitely does not deserve to be in the manuscript as it is not novel at all, which is also partially true for the Figure 2. They can be placed in the Supplement.
- No references have been given regarding IR peak assignment nor TG regions assignment.
- Very strange small steps at the TG figure exist for the purple curve, where does this comes from?
- Does the grafting “anomaly” at 50/50 ratio fit well with the rest of the date, especially SEM images?
- The authors of this study are comparing TG of the starch (but not AM) to the gel (that has more than 50 wt% of AM) identifying three more interesting weight loss regions, explaining them as a water evaporation, degradation of starch and degradation of AM. AM TG needs to be displayed in order to claim this. There could be potentially huge overlap between these regions that would make this assignment wrong. AM TG has to be added to the current graph for the comparison purposes.
Author Response
Dear reviewer
We have given a detailed answer to your question, please review it.

Reviewer 2 Report
- Line 41, seems to have an extra parenthesis or missing one.
- Line 247, Fig 4. Space or semi-colon is missing in “TGdegradation”
- Figures 3, 7 and 8, need to be presented with adequate statistical analysis.
- There is a lack of ultrasound heating background, why did this kind of methodology is used to generate the hydro-gel. What did the authors expect to obtain by using energy coming from an ultrasound source instead of a traditional thermal process?.
- The correct term is Sonochemical assisted synthesis and the real energy transfer comes from the implosion of the bubbles. So please explain the reason of heat the ultrasound bath? If the sonochemistry per se increases the temperature of the solutions.
- If the authors used a regular ultrasound bath, must present the frequency, amplitude, or power of the equipment.
- As a suggestion, the authors must present the difference between using a regular ultrasound bath and a sonotrode and how it will or can impact the results presented in the MS.
Reviewer 3 Report
Luo and coworkers fabricated and characterized starch-based absorbent hydrogels via changing the composition of amylose and amylopectin. Authors showed that by systematically changing the ratio of
amylose/amylopectin, the microstructure of hydrogels was changed and affected their water absorbing ability. While this study does provide valuable clues for researchers in designing starch-based hydrogels, extensive editing, including but not limited to language, is required to improve the flow and readers understanding. The title spotted on ultrasound-assisted construction, which is sort of out of focus since the manuscript mainly discussed the influence of starch composition on hydrogel properties. The order of presentation also can be altered. Figure 6 and the corresponding paragraph can be displayed as the first section in resultsto familiarize readers about the fabrication mechanism and nomenclature. An additional
chemical scheme will inform readers about the details of grafting and crosslinking chemistries. For example, the crosslinker methylenebisacrylamide and the utilization of cerium ammonium nitrate and
sodium hydroxide for grafting were never mentioned in the result until the materials and method section. Without this information, a sudden jump from the introduction to the result of FT-IR seems a bit awkward.
Authors should also double check the definition of rate and ratio. Throughout the manuscript, the terms of water retention rate, grafting rate, swelling rate were frequently mentioned. However, “rate” refers to
quantities involving “time.” Since none of the physical quantities in equations 1-3 is time-related, proper terms such as efficiency or ratio should be used. The experiment designs are fine but the presentation
and discussion can be improved. With the amount of editing that will involve, I recommend this manuscript to be published after major revision.
Please see other point-to-point comments below:
1. Page 3, Figure 1 and 2.1.1: To keep the nomenclature consistent, S-A gel in this section can be referred as AP0 sample. There is a mismatch between Figure 1 and line 90-93, as the peak of 3440 and 2930 are labelled for amylopectin in figure and amylose in the text.
2. Page 4, 2.1.2: section 2.1.3 should be displayed before 2.1.2 if the discussion of grafting ratio is intended on line 115-118 and 128-131.
3. Page 5, Figure 3: The x-axis “branched” is never defined in the text. It will be better to use ex. amylopectin wt% instead.
4. Page 5, line 137-139: The statement in this sentence lacks experimental support based on Figure 3.
5. Page 5, line 143: The meaning of “the change rule” is not clear.
6. Page 6, line 157-158: Since the degradation of starch gel occurred at 250-300oC and 350-450oC,which sandwiches the decomposition of pure starch at 300-350oC, additional evidence should be provided to prove that 250-300oC belonged to the degradation of starch and not AM.
7. Page 6, line 163-163: If starch gel started to degrade at 250oC, which is lower than 300oC of pure starch, why the thermal stability of starch gel is better than starch alone?
8. Page 8, line 218: The viscosity of the system was not discussed before this sentence. A bit more information will help readers understand the relationship between the extension of amylose and the increase in viscosity.
9. Page 9, line 235-238: It is probably me. But I had a hard time understanding these sentences. Since the swelling ratio reached maximum at 70% what is the sentence 235-236 referring to?
The swelling ratio decreased with reducing amylopectin content before 70%. AP50 also followed the trend. Where is the discussion of crystallinity coming from?
10. Page 11, line 280: Please double check the cross-reference of figures. Figure 1 is currently the FT-IR spectra.
Author Response

(The authors gave the same response as above.)

Round 2
Reviewer 2 Report
I don´t have any further comments. It seems like the authors respond to the reviewers' comments.
Author Response
Dear reviewer
According to your suggestion, we have revised part of the grammar of the manuscript. Thank you for your review and have a pleasant workday.
Yours sincerely
Reviewer 3 Report
Comments attached
As a reviewer, I appreciate authors efforts to edit the manuscript accordingly based on the point-to-point comments. I do think that there are technical merits in this article for researchers to link the starch composition to its hydrogel properties. Below are some additional comments and comments breaking down from my previous note.
1. The title may be re-focus on the starch composition influence on the hydrogel properties instead of “ultrasound-assisted construction.” This might catch more readers attention.
2. Flow of the manuscript. Figure 6 and the corresponding paragraph can be displayed as the first section to familiarize readers about the fabrication mechanism and nomenclature. An additional chemical scheme will inform readers about the details of grafting and crosslinking chemistries.
For example, the crosslinker methylenebisacrylamide and the utilization of cerium ammonium nitrate and sodium hydroxide for grafting were never mentioned in the result until the materials and method section.
I appreciate authors change to S-A gel to AP0 in 2.1.1. But readers do not know the definition of AP0 (or AP) on line 86 yet.
3. The version of the manuscript seems not in consistent with the one in authors response. In the revised manuscript, Figure 2 and 7 are in “branched” instead of “amylopectin wt%.”
4. On line 113, To describe the trend, something like “the grafting ratio of the starch hydrogels roughly follows its crystallinity” may work out better.
5. Regarding the viscosity, pointing out that amylopectin is the major buildup for viscosity will ease readers understanding.
6. There is a typo on line 217. It should be the relative content of “amylose” instead of amylopectin.

Author Response
Dear reviewer
We answered your questions based on your suggestions and uploaded them as attachments.
